# Molecular Simulations and Network Analyses of Surface/Interface Effects in Epoxy Resins: How Bonding Adapts to Boundary Conditions

**DOI:** 10.3390/polym14194069

**Published:** 2022-09-28

**Authors:** Julian Konrad, Paolo Moretti, Dirk Zahn

**Affiliations:** 1Lehrstuhl für Theoretische Chemie/Computer Chemie Centrum, Friedrich-Alexander Universität Erlangen-Nürnberg, Nägelsbachstraße 25, 91052 Erlangen, Germany; 2Institute of Materials Simulation, Friedrich-Alexander-Universität Erlangen-Nürnberg, Dr.-Mack-Str. 77, 90762 Fürth, Germany

**Keywords:** composites, molecular dynamics, curing, interfaces

## Abstract

In this study, we unravel the atomic structure of a covalent resin near boundaries such as surfaces and composite constituents. For this, a molecular simulation analysis of epoxy resin hardening under various boundary conditions was performed. On the atomic level of detail, molecular dynamics simulations were employed to study crosslinking reactions and self-organization of the polymer network within nm scale slab models. The resulting structures were then coarsened into a graph theoretical description for connectivity analysis of the nodes and combined with characterization of the node-to-node vector orientation. On this basis, we show that the local bonding of epoxy resins near interfaces tends to avoid under-coordinated linker sites. For both epoxy–vacuum surface models and epoxy–silica/epoxy cellulose interfaces, we find almost fully cured polymer networks. These feature a local increase in network linking lateral to the surface/interface, rather than the dangling of unreacted epoxy groups. Consequently, interface tension is low (as compared to the work of separating bulk epoxy), and the reactivity of the resin surface appears negligible.

## 1. Introduction

Epoxy polymer-based composites are of increasing importance for many technological developments aiming at the combination of low syntheses costs and tailor-made material properties. While the thermosetting polymer offers excellent processing characteristics, the mechanical properties of pure epoxy resins are unsuited for most applications. This motivated concepts of polymer reinforcement, particularly by the embedding of fibers or (nano-) particles to increase toughness in a controlled manner. The range of such composite formulations is immense, and combinations of favorable effects are often employed. For example, toughening of epoxy resins by particle insertion may also provide better resilience to abrasion. In parallel, the embedding of fibers may account for improvements in the characteristics of wear and fracture [1,2,3].

To name just one example from the vast number of different devices made of toughened epoxy resins, wet clutches are typically based on epoxy composites including both particles (silica, clay, etc.) and fibers (cellulose, aramid, etc.). The production process is akin to that of paper, entailing soaking of the fibers by water–particle mixtures, followed by pressing and drying of sheets or discs. Once dried, this “base paper” is then soaked by epoxy and hardener species, leading to the curing of the resin within a matrix of fibers and particles [4,5,6]. The enormous industrial relevance led to extensive engineering of constituents and processing parameters. Parallel to this, atomic force and electron microscopy improved our understanding of the µm scale structure [6].

For the understanding of such materials at the atomic/molecular scale, however, we fall short of experimental evidence. In turn, molecular simulation techniques offer in-depth insights into the manifold of atomic interactions accounting for the nature of the resin and its interplay with the composite constituents. This, however, requires careful preparation of the underlying simulation models, since basically no a priori knowledge of the polymer network structure is available apart from the (almost 100%) degree of curing observed for industrial epoxies. We argue that the most reliable approach to model polymer network structures is to simulate the actual formation process. For this purpose, we combined a reactive molecular mechanics model with molecular dynamics simulation protocols, thus following the evolution of chemical bonding and structural relaxation during the curing process [7,8]. On this basis, realistic network models of atomic resolution featuring bisphenol-F-diglycidyl ether (BFDGE) linked by 4,6-diethyl-2-methylbenzene-1,3-diamine (DETDA) were obtained [8]. The resulting polymer model was demonstrated to reproduce a series of bulk properties in excellent agreement with the experiment [9,10], including the degree of curing (99%), elastic properties, yield and ultimate stress [8,11].

In the present work, we transfer these simulation protocols to heterogeneous systems, namely slab models mimicking epoxy surfaces and interfaces. In line with the composite applications described above, the investigated interfaces are focused on epoxy contacts to silica and cellulose. For all of these systems, we shall revisit the curing process such that epoxy network formation is elucidated in the presence of the corresponding boundary condition. On this basis, unprejudiced insights into the local nature of polymer network arrangement near surfaces and interfaces may be achieved from direct comparison to the bulk resin.

## 2. Simulation Details

The epoxy matrix for each model (bulk, layer, interfaces with silica and cellulose slabs) was created from a 3D periodic simulation box featuring 1024 EPON and 512 DETDA molecules (Figure 1). Atomic interactions are treated by a reactive molecular mechanics model as fully adopted from our earlier study of bulk epoxy resin hardening [8]. Accordingly, the OPLS-AA forcefield is used in combination with additional short-range potentials that mimic the addition reaction of epoxy and amine groups (Figure 1). Moreover, we fully adopt the Monte Carlo-type approach [7] for exploring linking reactions from Ref. [8].

Unlike our previous study on bulk epoxy [8], here, we used non-cubic simulation boxes to better adjust the model setup to the envisaged analyses of surfaces and interfaces. The epoxy–vacuum model is based on a 2D periodic slab of 6.0 × 6.0 nm^2^. In turn, the silica–epoxy and cellulose-epoxy interfaces are described as sandwich models featuring 7.2 × 7.6 nm^2^ and 6.0 × 6.0 nm^2^ sized layer segments, respectively, in 3D periodic simulation boxes. In line with typical experiments, the curing is investigated at 460 K and 1 atm pressure [8]. This also applies to the epoxy surface model, as we temporarily introduced repulsive walls for restraining the slab by half harmonic walls with a force constant of 5 kcal mol^−1^nm^−2^. After completion of the curing procedure, the repulsive walls are removed and the epoxy–vacuum slabs were allowed to relax at zero pressure within a 100 ns run at 300 K.

In line with the epoxy forcefield, the interactions of cellulose were also described by the OPLS-AA model [12]. As a proxy to the layer-wise stacking of cellulose fibers (of typically disperse length distribution), our nm scale model system was built up with seven parallel strands of eleven β(1-4) linked D-glucose molecules forming an area of 36 nm^2^, which was stacked five times, resulting in slab of 1.8 nm thickness (Figure 2, left).

While the silica–epoxy interactions were also described by OPLS-AA, the silica–silica interactions are more accurately described by a tailor-made silica/silanol potential [13]. We adopted this forcefield—and the 2D periodic model of an amorphous silica slab with full surface hydroxylation—from Ref. [14]. On this basis, our 54.5 nm^2^ sized silica slabs of 1.8 nm thickness, as illustrated in Figure 2, feature 3024 Si, 5799 O^2−^ and 498 OH^-^ ions. 

For the van der Waals interactions, we use cut-off potentials with a distance delimiter of 1.2 nm, whereas the Coulomb interactions were treated by the damped-shifted force potential [15] using a cut-off distance of 1 nm and a damping factor of 0.1, as suggested by Fennel [16]. The molecular dynamics simulations were carried out with LAMMPS using a timestep of 0.5 fs and the built-in the Nosé–Hoover algorithm for maintaining constant temperature and pressure. To ensure decoupling of thermostat and barostat fluctuations, the corresponding relaxation times were chosen as 0.1 ps and 1.0 ps, respectively. Figure 2Illustrations of the cellulose (**left**) and silica (**right**) slab models. Finite cellulose fibers are connected by hydrogen bonds to form staggered layers with 2D periodic boundaries in the x and y directions. The amorphous silica slab (also using 2D periodic boundaries in x–y directions) features charge-neutral hydroxylated surfaces to avoid dangling O^2−^ ions, as adopted from Ref. [14].
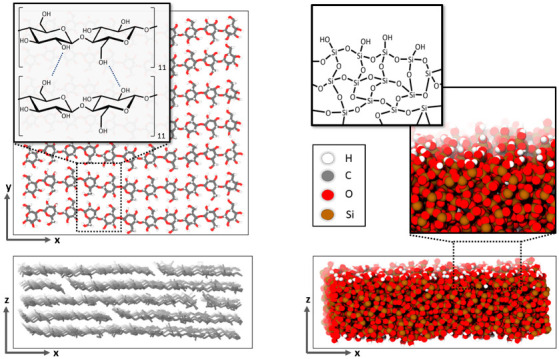



While molecular simulations are used for the modeling of linking reactions and self-organization of the epoxy network, once the curing process of the thermosetting resin is accomplished, coarsening into a description inspired by graph theory offers a more general understanding of connectivity. For this, we define the chemical linker species, DETDA, as network nodes, whereas the EPON molecules are considered as network edges. In the fully cured epoxy resin, i.e., crosslinking degree of η = 100%, each node is 4-fold coordinated (two addition reactions per –NH_2_ moiety) by edges. In turn, both ends of the edges (the two –COC moieties of EPON) will then be connected to the linkers. 

Methods of graph theory often focus on adjacency relationships between pairs of nodes, starting with the local concept of a node’s degree (its number of neighbors, or coordination), and moving to global quantities such as degree distributions and degree–degree correlations [17]. In our case, degree measures are directly reflected by the number of covalent bonds formed between DETDA and EPON. We count this for each of the N atoms per DETDA separately and distinguish single and twofold reaction of the amines, respectively. Apart from such “primary” information of node–edge coordination, it is helpful to analyze node–edge–node relations in terms of their structural alignment. For this reason, we chose to focus on edge orientations to monitor the local directionality (node–edge–node alignments) of the network. For every edge, the orientation is computed by defining a unit vector $\vec{v}$ connecting the two –COC moieties of the corresponding EPON (see also Figure 1). Next, the angle $\alpha_{}$ between $\vec{v}$ and the z-axis direction (i.e., the normal of the surface/interface models) is calculated. Moreover, the center of mass positions of the involved DEDTA and EPON species are used for marking the location of the network nodes and edges, respectively. This allows the discrimination of different regions within our analyses of network connectivity. On this basis, the network analyses were performed as a function of the distance from the center of mass of the epoxy slabs, taken along the surface/interface normal vector.

The statistical analysis of the angular occurrence profiles P(α) is devised in intervals along z. For this, normalizing by 1/sinα is applied, so that P(α) is flat when all angles are equally likely to occur (bulk behavior). To construct the graph, node–edge connectivity was identified by a simple distance threshold of 0.18 nm—which is generously larger than the length (0.14 nm at equilibrium) of the underlying N-C bonds, including twisting or elastic deformation. 

## 3. Results

To investigate the local structure of epoxy surfaces and interfaces in an unprejudiced manner, each of the three sandwich-type simulation boxes was subjected to in-depth curing analyses from independent molecular simulation runs. Rather than simply attaching blocks of different constituents, our composite models describe resins that were cured in the presence of specific surface/interface boundaries. To ensure full comparability of the sandwich models to the bulk resin, the modeling and simulation protocols are implemented in full analogy to our earlier study of bulk epoxy resin curing.

We hence start from stoichiometric mixtures of randomly arranged *non-reacting* (standard OPLS-AA models) DEDTA and EPON species [8]. In separate pre-equilibration runs, all sandwich models were subjected to 5 ns runs at 460 K and 1 atm, allowing the two constituents to relax with respect to each other and to the given boundary condition (repulsive wall, silica and cellulose slabs). Convergence was monitored by following the occurrence profiles of DEDTA and EPON center-of-mass taken along the interface normal. On this basis, we observed full mixing of the two constituents for all sandwich systems—both in the epoxy bulk and at the boundaries. 

Upon switching to our reactive forcefield (see Ref. [8] for details), a large number of linking reactions occur spontaneously—thus, a crosslinking degree of η~80% is already observed from 2 to 5 ns straightforward molecular dynamics simulation runs at 460 K and 1 atm. When monitoring the time-evolution of crosslinking, it is interesting to discriminate the interface/surface domain (which was taken as the outermost 1 nm along the normal vector of the upper and lower faces of the epoxy slab) and the bulk region (which was taken as the central 2 nm). While statistics taken for such local slices are limited, we still observe a clear trend for all three interface/surface models investigated. Indeed, Figure 3 indicates faster crosslinking in the bulk as compared to the epoxy–vacuum, epoxy–silica or epoxy–cellulose boundaries. Considering the 1 nm resolution of our analyses, the main cause of the lower rate of crosslinking near the interfaces is surely given by the lack of potential binding partners, which would amount to 50% for the outermost molecules, yet curing at the boundaries does not simply occur at half of the pace found for the bulk.

Upon increasing the degree of crosslinking, the viscosity of thermosetting polymer increases drastically and full curing is thus beyond the scope of direct molecular dynamics simulations. To enable further crosslinking, we therefore employ the Monte Carlo procedure reported in Ref. [8]. In full analogy to our earlier study, we thus select nearby reaction candidates and attempt linking by temporarily enforcing the epoxy bonding distance. Each linking step is followed by 10 ps relaxation runs that allow for (i) the separation of reactants in case of unfavorable energy, and (ii) the rearrangement of the forming polymer network to best accommodate the increasing level of curing. The latter type of relaxation also includes the possibility of dissociating previously formed bonds at any location in the polymer network. These Monte Carlo/molecular dynamics simulation runs are continued iteratively until convergence is concluded from the inability to further increase the degree of crosslinking (using a delimiter of 1000 failed attempts to end the iterations) [8]. To support this technical convergence criterion, we also calculated the mass density of the bulk epoxy domains. For both the three-dimensional bulk (taken from Ref. [8]) and all sandwich models, we find the mass density to change from 1.07 g/cm^3^ (before curing) to 1.16 g/cm^3^ (fully cured), thus suggesting convergence of the curing process also in terms of resin density.

On this basis, in each of three sandwich models, the epoxy resin of was cured within ~100 ns scale simulation runs, achieving crosslinking degrees of η = 96–98% in the bulk of the epoxy slabs (Figure 3, Table 1). Comparing these findings to the crosslinking degree of η = 99% for the bulk epoxy system reported in Ref. [8], we thus observe a reduction of 1–3%. Interestingly, the findings for the bulk epoxy domains comply with analyses of the local crosslinking degree within the outermost 1 nm surface/interface domains of the epoxy slabs performed for all three sandwich models (Figure 3).

Intuitively, one might expect a large number of dangling epoxy linker moieties at the boundaries of the resin at the end of the curing runs. However, Figure 3 indicates local values of η = 94% for the surface of the epoxy–vacuum slab, whereas 93% and 96% are found at the interfaces of silica and cellulose, respectively. Apart from the ~5% under-coordination of the DEDTA and EPON molecules at the resin boundaries, we also find that there is a broad distribution of unreacted moieties throughout the entire resin slabs of our models. Indeed, the surface slab (Figure 4a) and epoxy interfaces of cellulose (Figure 4b) or silica (Figure 4c) all displayed small content of unreacted DEDTA/EPON moieties, which are both distributed over the entire epoxy slabs of the sandwich models (Figure 5). This suggests that imposing geometric boundaries during the curing of the thermosetting resin implies a complex frustration of the formed polymer network. This frustration extends (at least) over several nm scales—possibly even the entire epoxy domain in our simulation boxes.

To demonstrate the interplay of crosslinking reaction energy and the mechanical stress in the thermosetting resin, we calculated the heat of polymerization, directly assessable from our reactive forcefield as the sum of all epoxy–epoxy interaction energy terms [8]. Along this line, we contrasted the heat of polymerization of the epoxy slab model to that of the bulk resin (comprising the same number of molecules). The negative difference ΔQ = Q_slab_ − Q_bulk_ indicates the disfavoring of epoxy–vapor boundary formation. For the sake of comparability, this difference in heat of polymerization ∆Q = −0.06 J/m^2^ is normalized per surface area (Table 1). To discriminate the role of unreacted linker/epoxy moieties for network deformation, it is educative to compare Q_slab_ to the expected heat of polymerization of a bulk epoxy model (with the same number of molecules), but at a <η> = 96% degree of crosslinking (Table 1), namely Q_bulk_ × 96/99. On this basis, we can characterize ΔQ = ΔQ_reaction_ + ΔQ_network_ with ΔQ_reaction_ = −0.17 J/m^2^ and ΔQ_network_ = 0.11 J/m^2^. In other terms, the unreacted linkers found in the slab model enable significant rearrangement of the polymer network (as compared to the 3D periodic bulk), which leads to mechanical stress reduction that compensates about two-thirds of the loss in formation energy ΔQ_reaction_ stemming from the 3% lack of linking reactions.

To elucidate the change in the polymer network induced by the epoxy surface, we furthermore analyzed the orientation of the EPON entities with respect to the surface/interface normal vector. In Figure 6, occurrence profiles P(α) of the EPON angle orientation are shown for various regions within the epoxy layers of the surface slab and the interfaces of silica and cellulose, respectively. For this, we discriminate surface/interface regions from the slab center by cutting slices as functions of the distance z (taken along the direction of the normal vector) from the center of the slab. The choice of the underlying delimiters is somewhat arbitrary; however, our selection was motivated by the density profiles ρ(z) of epoxy nodes, as illustrated in Figure 5. Thus, the onset of the decline in ρ(z) near the epoxy boundaries is assumed as the beginning of the surface/interface domains, whilst 5 and 1 nm sized intervals in the center of the epoxy slab are taken as coarse and fine delimiters of the bulk regions. Using either of these definitions of surface/interface and bulk domains, we do not find pronounced differences in the ratio of fully coordinated and under-coordinated network nodes. However, the local orientation of the EPON moieties indeed shows distinct preferences of lateral orientation near the surface/interface of the epoxy slabs (Figure 6). In more qualitative terms, the polymer network arranges its linking in such a manner that dangling moieties of unreacted DEDTA/EPON species at the surface/interface are largely avoided. This applies to all three boundary scenarios investigated. 

To characterize the two interface models quantitatively, we first evaluated the heat of polymerization in analogy to the epoxy–vacuum system. In comparison to the surface model, the epoxy–silica and epoxy–cellulose interfaces show even stronger reductions ∆Q in the heat of polymerization per boundary area. However, the observed loss in epoxy–epoxy reaction energy was found to be over-compensated by epoxy–silica and epoxy–cellulose interactions, namely hydrogen bonding at the interface. This is reflected by the overall interface energy σ, which we define as the difference in total energy (per interface area) of the sandwich model as compared to the bulk epoxy resin and isolated silica/cellulose slabs. On this basis, we find negative (thus exothermic) interface formation energy for both silica and cellulose incorporation into the epoxy resin (Table 1).

While the energetics discussed above refer to the processes of resin and resin-based composite formation, the work of separation is of key importance for mechanical characterization. In our recent study on bulk epoxy deformation and fracture [8], the fracture process was identified as a complex combination of viscous and plastic deformation in parallel to gradual network dissociation. Unsurprisingly, the frayed structures of the fragments resulting from tensile pulling differ quite drastically from the epoxy slabs discussed above. This is also reflected by the work of separation, which amounts to 1 J/m^2^ for the bulk model of Ref. [8]. In contrast to this, the separation of the epoxy–silica and epoxy–cellulose interfaces occurs as well-defined dissociation of the distinct composite constituents. The underlying loss of hydrogen-bonded contacts gives rise to much lower work of separation as compared to bulk epoxy fracture (Table 1). Separation of the sandwich models implies only very local reorganization of the resulting epoxy and silica/cellulose fragments. Nevertheless, the work of separation—calculated as the integral over the restoring force as a function of displacement—amounts to less than −2 times the interface energies σ. In other terms, upon dissociation of the epoxy–silica and epoxy–cellulose interfaces, more than half of the work of separation is transferred into heat, namely the heat of re-forming hydrogen bonds among the surface ions of the silica slab and the outermost molecules of the cellulose model, respectively.

## 4. Discussion and Conclusions

In the past few years, molecular simulations achieved the in-depth analyses of the structure of polymer networks, including thermosetting resins [18,19,20,21,22]. This was enabled by studying the curing process itself, hence providing unprejudiced structure models of polymers for which little a priori knowledge of the network was available. While we lack experimental data on the atomic structure of epoxy networks, confirmation of the simulation models was provided in recent studies [7,8,11] on the basis of macroscopic properties assessable from experiments such as density, glass transition temperature, mechanical properties, degree of curing and the heat of polymerization. This suggests that our molecular simulation techniques can readily tackle the structure and properties of bulk epoxy resins, and we conclude analogous robustness for describing epoxy surfaces and interfaces. 

In the present study, we unravel the atomic structure of DEDTA/EPON thermosetting polymers next to three rather different boundaries to the resin. Inspection of the curing reactions, however, led to rather similar findings regarding the degree of linking and the arrangement of links between the nodes of the polymer networks. For both the epoxy surface and the interface models related to epoxy–cellulose and epoxy–silica composites, we found the polymer network to arrange parallel to the boundaries. The implication for the nature of epoxy-based composites is illustrated in Figure 7. The scheme discriminates two different types of embedding constituents in polymer composites—of which the arrangement of network links “around” the inserted species appears most suitable, despite the need for several nm scale reorganizations of the polymer phase. In turn, dangling entities of unreacted EPON and DEDTA species at the interfaces seem to be disfavored quite generally.

Our findings underpin the importance of explicitly simulating the curing process in order to achieve realistic molecular simulation models of epoxy-based composites. Likewise, the proper account of nano-fibers and nano-scale layers of epoxy resins calls for the careful analysis of polymerization reactions in the presence of appropriate boundary conditions [23,24,25,26]. We argue that our Monte Carlo-type approach of crosslink attempting, relaxing and dissociating if overstressed is particularly suited for this purpose [8], because artificial accumulation of local stress from poorly placed crosslinks is avoided automatically. 

While the present study was focused on flat boundary scenarios, we caution that studies of epoxy composites featuring particles of small dimensions (e.g., less than ten times the 2–3 nm scale of the observed rearrangements in the polymer phase) may call for explicitly considering curvature effects. This also applies to the consideration of chemical modification of particle surfaces, such as the introduction to amine moieties to achieve linker-type functionalization of silica particles that enable covalent incorporation into the epoxy network [26,27,28,29].

## Figures and Tables

**Figure 1 polymers-14-04069-f001:**
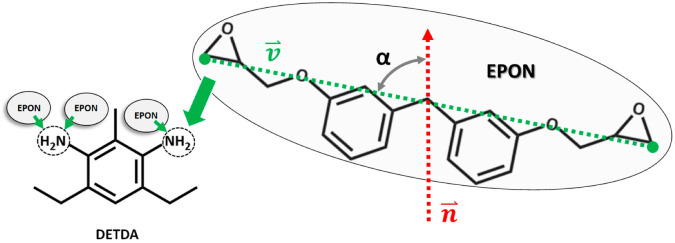
DETDA (**left**) used as linkers to connect EPON molecules (**right**). In the ideal resin (curing degree of η = 100%), all hydrogen atoms of the amine groups participate in linking reactions. As a consequence, DETDA is fourfold coordinated by epoxy moieties (which in turn are twofold coordinated by DEDTA). To interpret the resulting network structure by graph theory, we identify DETDA as network nodes and EPON as edges. The alignment of EPON molecules is further characterized by the orientation of the vector $\vec{v}$ connecting the terminal carbon atoms of the –COC moieties (highlighted in green). From this, network alignment with respect to dedicated directions—such as the normal $\vec{n}$ of surfaces/interfaces—is characterized.

**Figure 3 polymers-14-04069-f003:**
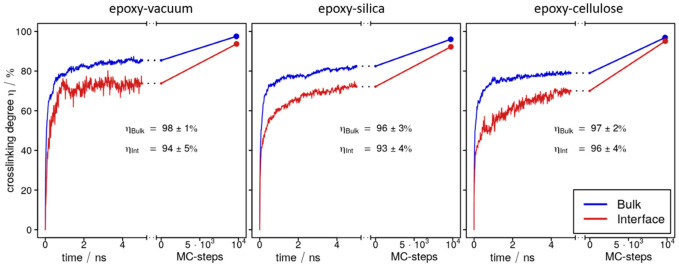
Evolution of the crosslinking degree as obtained from curing simulations for epoxy interfaces of vacuum (**left**), silica (**center**) and cellulose (**right**). Data are sampled for the bulk (blue curves) and surface/interface (red) domains separately. Each system is first explored from direct molecular dynamics simulation runs of 5 ns, and Monte Carlo (MC) steps are then employed to boost crosslinking towards 100%. To this end, convergence of the resin curing is achieved at the ending point of the MC simulations, i.e., the steep increase in failed MC moves (>1000) for the final crosslinking attempt. The error margins are taken as the differences observed when sampling upper/lower 1 nm slices independently.

**Figure 4 polymers-14-04069-f004:**
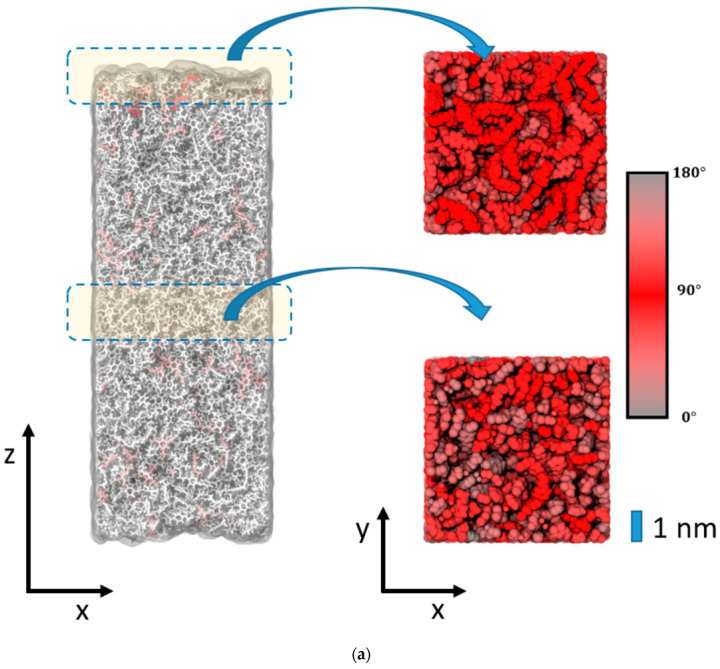
(**a**) Illustration of the epoxy slab model after convergence of the curing reactions (η = 96%). Left: side view of the epoxy–vacuum interface. Fully reacted DEDTA and EPON moieties are shown in gray. In turn, coloring by pale red to dark red indicates under-coordination. Note the homogeneous occurrence of under-coordinated linkers, which shows no significant increase near the boundaries of the epoxy resin phases. Right: top view of the epoxy surface and a slice cut through the bulk resin. Here, red color is used to highlight the EPON tilt angle with respect to the surface normal (*z*-axis). (**b**) Analogous to (**a**), but illustrating the epoxy–silica sandwich model. The picture on the left is based on a 1 × 1 × 2 super cell of the actual simulation system. All atoms of the silica slab are shown in brown color. (**c**) Analogous to (**a**), but illustrating the epoxy–cellulose sandwich model. The picture on the left is based on a 1 × 1 × 2 super cell of the actual simulation system. All atoms of the cellulose slab are shown in green color.

**Figure 5 polymers-14-04069-f005:**
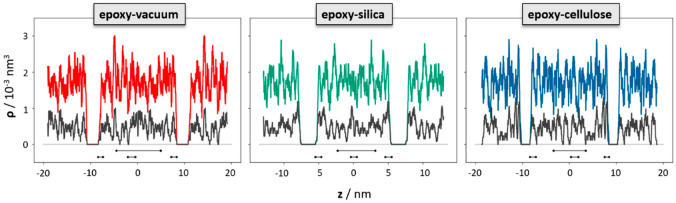
Density profiles of the polymer network nodes as functions of the z-coordinate taken along the surface/interface normal. While the colored curves indicate the overall density of nodes in the epoxy–vacuum (red), epoxy–silica (green) and epoxy–cellulose (blue) sandwich models, respectively, the curves shown in black indicate the occurrence of under-coordinated nodes (missing at least 1 link) in the corresponding systems. The occurrence of under-coordinated nodes shows fluctuations by ±25% over the entire epoxy phase. Within this level of accuracy, we do not observe significant differences near the boundaries. The black bars indicate the z-intervals used to provide local statistics of polymer network alignment in the upper and lower surface/interface regions and the bulk domains, respectively.

**Figure 6 polymers-14-04069-f006:**
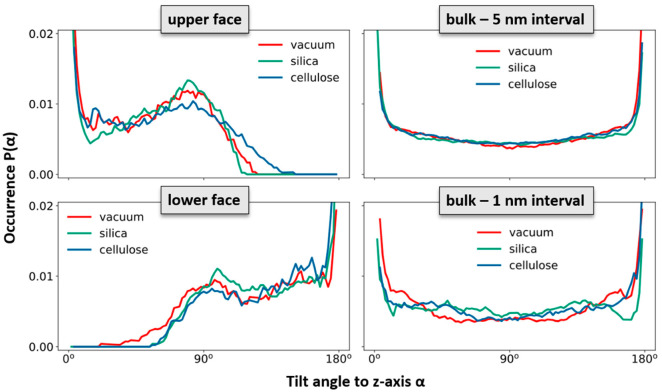
Occurrence profiles P(α) of the orientation of EPON moieties in the polymer network. The angle α refers to the tilting of the network edges with respect to the normal vector $\vec{n}$ of the surfaces/interfaces. Because of the symmetric nature of the EPON molecule, the sign of its orientation vector $\pm \;\vec{v}$ is ambiguous. We therefore used the location of the connected nodes to define the upper and lower ends of the network edges. To account for the reduction in 3D vectors to their tilting angle with respect to a single direction, all occurrence statistics were normalized by 1/sin(α). This provides flat occurrence profiles for random orientation of the edges in the epoxy bulk. Because of this normalization, data quality is proportional to sin(α), and fluctuations near α = 0° and α = 180° have no physical origin. In turn, the observed signatures near α = 90° are statistically significant. The statistics collected for the bulk phase (right) were performed for sampling of 1 and 5 nm sized intervals to illustrate the error margins from statistical fluctuations. To define the surface/interface regions of all simulation systems, 1 nm sized sampling intervals are used.

**Figure 7 polymers-14-04069-f007:**
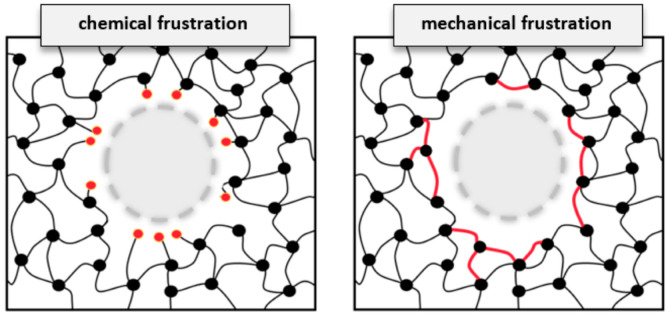
Schematic illustration of particle embedding into the covalently bonded network of a thermosetting polymer. By the example of epoxy resins, we contrasted two conceptual types of interface formation. This includes (**left**) the possibility of dangling network nodes at the interface, thus featuring chemical frustration from under-coordination, but leaving the bulk network largely unchanged. In turn, (**right**) alignment of polymer linking “around” the imposed boundaries leads to better curing at the interface, but implies larger-scale mechanical frustration of the overall polymer network.

**Table 1 polymers-14-04069-t001:** Chemical and mechanical data collected for the epoxy surface model and interfaces of silica and cellulose, along with comparison to the bulk (where applicable). Averages of the degree of crosslinking <η> and the heat of polymerization ΔQ_polymerization_ are sampled over the entire simulation models. For the epoxy–vacuum system, the surface energy σ_interface_ = −ΔQ_polymerization_, whereas epoxy interfaces of silica and cellulose layers feature favorable hydrogen bonding between the composite constituents that significantly lower the interface energy. The work of separation W_separation_ of the interface systems was found as about 20% of that identified for the bulk epoxy model (adopted from Ref. [8]), indicating the epoxy–silica and epoxy–cellulose contacts as preferred nucleation sites for cavitation and fracture processes in the corresponding composite resins. Block-wise separation of the upper and lower epoxy–silica and epoxy–cellulose interfaces was studied in separate runs. The reported data of W_separation_ refer to the averages obtained, whereas the deviation of the two types of runs is less than 0.01 J/m^2^. This value is suggested as an error margin for all surface/interface energy calculations.

	Epoxy–Vacuum	Epoxy–Silica	Epoxy–Cellulose	Bulk (Ref. [8])
〈η〉/%	96	96	96	99
$\;\Delta Q_{polymerisation}\;/\frac{J}{m^{2}}$	−0.06	−0.08	−0.1	0 (reference)
$\sigma_{interface}\;/\;\frac{J}{m^{2}}$	0.06	−0.08	−0.03	-
$W_{separation}\;/\frac{J}{m^{2}}$	-	0.20	0.17	1.0

## Data Availability

Not applicable.

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
