# Peer review of "Molecular Simulations and Network Analyses of Surface/Interface Effects in Epoxy Resins: How Bonding Adapts to Boundary Conditions"

_polymers, 2022, doi:10.3390/polym14194069_

Round 1

Reviewer 1 Report

The present work is a continuation of the research presented in the author’s previous paper (ACS Polym. Au 2021, 1, 165−174). Based this experience, the authors simulated the curing of epoxy resin at the interface with vacuum, silica, and cellulose. The work is well performed, the results are appropriately presented and discussed. I will recommend it for publications with minor amendments.

Some remarks:

-          I suggest changing “simulation cell” to “simulation box”, which is less confusing.

-          Page 3, the sentence “Unlike our previous study, in what follows we however use non-cubic simulation cells to better adjust the model setups to the envisaged analyses of surfaces and interfaces” is not clear. The reader will not know which 'previous study' the authors are referring to. The sentence should be rewritten.

-          Page 4, the descrition “For this reason, we choose to focus on edge orientations. For every edge ?, connecting nodes ? and ?, we identify the unit vector ?⃗ originating in ? and directed along ?. We define the orientation ??? of edge ? at node ? as the angle between ?⃗ and the ? axis direction (i.e. the normal of the surface/interface models). Similarly, we compute ??? at node ?, and notice that in our geometry the two angles are supplementary. Since we have defined angles as node-wise quantities, we can now accurately measure the prevalence of certain tilt angles at interfaces and contrast that to bulk behavior, something that would have been impossible measuring angles, e.g., at the edge center. To this end, we compute the angular occurrence profiles ?(?) for a given interval in z, by counting the occurrences of all an-gles in that interval and normalizing by 1/sin?, so that ?(?) is flat when all angles are equally likely to occur (bulk behavior)” is completely incomprehensible. The authors are suggested to show a drawing explaining what is “edge ?” and “the orientation ??? of edge ? at node ? as the angle between ?⃗”.

-          Page 4, why was “distance threshold (0.18 nm)” chosen? Is this the length of a C-N bond? Each bond has a well-defined geometry and is therefore formed with a close alignment of the atoms forming the bond (orientation factor). In addition to the distance criterion, did the authors also consider the geometry of the bond (criteria for the relevant angles)?

-          Page 6, “models the epoxy resin of was cured with”?

-          No cation for Figure 6.

-          Although the authors wrote “While we lack experimental data on the atomic structure of epoxy networks, confirmation of the simulation models was provided on the basis of density, glass transition temperature, mechanical properties, degree of curing and the heat of polymerization” no experimental data for comparison were provided in the manuscript. An in-depth discussion of the results obtained based on the experimental data is advisable. Also, the conclusions should be more sharpened.

Author Response

Thanks for the constructive hints.

- We changed  “simulation cell” to “simulation box” at all places.

- Page 3, the sentence “Unlike our previous study, in what follows we however use non-cubic simulation cells to better adjust the model setups to the envisaged analyses of surfaces and interfaces” was rewritten to

"Unlike our previous study on bulk epoxy [8], in what follows.."

  • Page 4, the descrition “For this reason, we choose to focus on edge orientations. For every edge ?, connecting nodes ? and ?, we identify the unit vector ?⃗ originating in ? and directed along ?. We define the orientation ??? of edge ? at node ? as the angle between ?⃗ and the ? axis direction (i.e. the normal of the surface/interface models). Similarly, we compute ??? at node ?, and notice that in our geometry the two angles are supplementary. Since we have defined angles as node-wise quantities, we can now accurately measure the prevalence of certain tilt angles at interfaces and contrast that to bulk behavior, something that would have been impossible measuring angles, e.g., at the edge center. To this end, we compute the angular occurrence profiles ?(?) for a given interval in z, by counting the occurrences of all an-gles in that interval and normalizing by 1/sin?, so that ?(?) is flat when all angles are equally likely to occur (bulk behavior)” is completely incomprehensible. The authors are suggested to show a drawing explaining what is “edge ?” and “the orientation ??? of edge ? at node ? as the angle between ?⃗”.

This text is now rewritten (much) more clearly and supported by figure 1.

  •  Page 4, why was “distance threshold (0.18 nm)” chosen? Is this the length of a C-N bond? Each bond has a well-defined geometry and is therefore formed with a close alignment of the atoms forming the bond (orientation factor). In addition to the distance criterion, did the authors also consider the geometry of the bond (criteria for the relevant angles)?

This was written to "To construct the graph, node-edge connectivity was identified by a simple distance threshold of 0.18 nm – which is generously larger than the length (0.14 nm at equilibrium) of the underlying N-C bonds, including twisting or elastic deformation."

-Page 6, “models the epoxy resin of was cured with”?  - within

  • Caption of fig.6 was reformatted

-conclusion:

we clarified the discussion of experimental data. Moreover, we point out that the main findings of the present work are atomic scale insights that explain the more macropscopic data collected from experiments.

Reviewer 2 Report

This manuscript reports a study of molecular simulations and network analyses of surface/Interface effects. The author proposed interesting and comprehensive analysis about the interface. Overall, this report presents a systematic work. However, this reviewer felt that the author may want to highlight the importance of your work in the abstract and further organize your conclusions.

Specifically, the abstract better start with a brief description of background, e.g. limitations/challenges/the importance of your work, followed by the description of your work and highlight the most important conclusions

In your conclusion, it is more important to summarize your work, method, and important conclusions. The illustration looks nice, but it might be more suitable to give it a thorough discussion in Results and then highlight it again in the conclusion. 

Author Response

Thanks for the usefull hints.

We rewrote the abstract and conclusions to improve assessability of our work.

We feel that the illustration should remain part of the 'conclusion and discussion' section. This avoids re-iterating the underlying discussion.

Reviewer 3 Report

The manuscript by Konrad et al. is related to simulations comprising atomistic molecular dynamics and Monte-Carlo of epoxy resin hardening near a vacuum and two different interfaces (silica or cellulose). This research is well-written and interesting. The results obtained support the conclusions. The authors could add more details to the manuscript by responding to some issues:

- How many initial samples are investigated for each system considered?

- What are the error bars of the values in Table 1?

Author Response

Thanks for the constructive reviewer comments.

Because of the already quite substantial computational demand, each we could only investigate a single model for each interface system. In turn, statistical sampling is determined by the size of the interface area - and by the availability of two interfaces regions in each model system. By contrasting upper and lower surfaces/interfaces of the resin, we already provided error margins to the local analyses of the degree of cross-linkinging in figure 3.

This concept is now also applied to the work of separation reported in table 1. By separating the system a) at the upper and b) the lower interface, we computed two individual energies and suggest the deviation from the average as error margins to all surface/interface energy calculations.

Reviewer 4 Report

Review of “Molecular simulations and network analyses of surface/interface effects in epoxy resins: How bonding adapts to boundary conditions”

By Konrad et al.

The paper presents a study using dynamics molecular simulations to examine the interface features of epoxy-based composites under various boundaries. The authors examined the local bonding near interfaces of different epoxy systems: epoxy-vacuum surface models and epoxy-silica / epoxy cellulose interfaces.

The paper is nicely written and the linguistic style is pretty good. The different stages of the proposed methodology are well exposed and seem comprehensively described. The results and the provided examples seem adequate. The related discussions are very interesting.

The paper deserves in my opinion to be part of the state of the art. I recommend accepting this manuscript for publication in Polymers after minor revisions. The two main issues are as follows:

1) The present study examines filled epoxy systems. Can the author add a discussion about the filler effect in terms of concentration and size? It would be interesting in future works to integrate this critical issue that remains an ambitious challenge.

2) A drawback here is that the predictions have not been verified. The properties predicted by the simulations are not compared to other simulation studies. Can the authors compare some of their results with existing experimental findings, at least qualitatively?

3) You can introduce a limitation section in order to highlight the main drawbacks of the modeling and of the present computations. It is not a strong criticism, but you need to mention this aspect in a good place in the paper (may be before the conclusion). Can you discuss this point and provides some propositions for the improvements of the predictions?

Author Response

Thanks for the constructive hits.

Our work is focused on a rather fundamental issue, namely perfectly 2D stacks of parallel layers. There is not explicit experimental analog to compare with. However, our study forms the basis for larger scale modelling of composites - which will then include concentration, shape and size of particles/fibers. Thus, the points that you envision are in our mind, but I feel it would be overambitious to discuss them at the current stage.